# *Plasmodium*, the *Apicomplexa* Outlier When It Comes to Protein Synthesis

**DOI:** 10.3390/biom14010046

**Published:** 2023-12-29

**Authors:** José R. Jaramillo Ponce, Magali Frugier

**Affiliations:** Université de Strasbourg, CNRS, Architecture et Réactivité de l’ARN, UPR 9002, F-67084 Strasbourg, France; jjaramillo@unistra.fr

**Keywords:** tRNA, amino acid usage, AlphaFold2 modeling, multi-synthetase complexes, translational control

## Abstract

*Plasmodium* is an obligate intracellular parasite that has numerous interactions with different hosts during its elaborate life cycle. This is also the case for the other parasites belonging to the same phylum *Apicomplexa*. In this study, we bioinformatically identified the components of the multi-synthetase complexes (MSCs) of several *Apicomplexa* parasites and modelled their assembly using AlphaFold2. It appears that none of these MSCs resemble the two MSCs that we have identified and characterized in *Plasmodium*. Indeed, tRip, the central protein involved in the association of the two *Plasmodium* MSCs is different from its homologues, suggesting also that the tRip-dependent import of exogenous tRNAs is not conserved in other apicomplexan parasites. Based on this observation, we searched for obvious differences that could explain the singularity of *Plasmodium* protein synthesis by comparing tRNA genes and amino acid usage in the different genomes. We noted a contradiction between the large number of asparagine residues used in *Plasmodium* proteomes and the single gene encoding the tRNA that inserts them into proteins. This observation remains true for all the *Plasmodia* strains studied, even those that do not contain long asparagine homorepeats.

## 1. Introduction

*Apicomplexa* is a group of obligate intracellular parasites with more than 6000 described species [1]. Many of these parasites are important pathogens in humans. The *Apicomplexa* phylum includes (i) *Plasmodium*, the parasite responsible for malaria, a mosquito-borne disease that is potentially deadly [2,3]; (ii) *Toxoplasma gondii*, a source of toxoplasmosis associated with congenital neurological birth defects (for example, encephalitis and ocular disease) [4,5]; and (iii) *Cryptosporidia*, which cause opportunistic infections in immunosuppressed patients transmitted via contaminated food or water [6,7]. Further, numerous infectious diseases in wild and domesticated animals are caused by members of the apicomplexan genera *Babesia*, *Theileria*, and *Neospora*.

Apicomplexan parasites have an unusual biology as intracellular eukaryotes growing inside another eukaryotic cell, which differentiates them from other pathogens. One of the most fascinating aspects that defines the biology of *Apicomplexa* is their ability to manipulate their host cells [8]. During infection, the parasites cause changes not only in the signaling or evasion of host immunity but also in manipulating host cells to suit their own needs, such as supplying nutrients and molecular building blocks (amino acids, nucleotides, glucose, etc.) to the parasite. Among the many strategies put in place to allow optimal development in the host cell, a singular example is provided by *Plasmodium*, which is characterized by the presence of tRip (tRNA import protein), a unique tRNA transporter that also participates in the formation of parasite multi-aminoacyl-tRNA synthetases complexes (MSCs).

Aminoacyl-tRNA synthetases (aaRSs) are essential enzymes that ensure the attachment of a specific amino acid to its cognate tRNAs [9]. In eukaryotes, a subset of cytosolic aaRSs are organized into MSCs built on multifunctional aaRS-interacting proteins (AIMPs) (reviewed in [10,11,12]). Despite their diversity, the assembly of these MSCs in protozoa follows a dominant strategy involving GST-like domains. Such domains are found to be essentially fused to AIMPs, aaRSs, and elongation factor subunits, and interact with each other via two well-identified binding surfaces called interfaces 1 and 2. In *Plasmodium*, tRip is an AIMP like no other: it is an integral membrane protein anchored in the plasma membrane of the parasite [13,14]; it homodimerizes using an alternative interface, named 1′ [13,15,16]; and it allows the formation of two MSCs characterized by two different GST-like domain organizations [13]. Most importantly, tRip is involved in the import of exogenous (host) tRNAs into the parasite [14] and its deletion leads to low infectivity and translational efficiency, which especially affects asparagine-rich proteins [17]. To date, it has not been demonstrated that *Apicomplexa* other than *Plasmodium* import and use host tRNAs.

In this study, we investigated the possibility for other parasites of the *Apicomplexa* phylum to form MSCs equivalent to those found in *Plasmodium*. To reconstruct the different MSCs’ architectures, sequence analysis and AlphaFold2 modeling were combined. This work was performed on four other genera apart from *Plamodiidaes* by selecting several parasites with complete and annotated genomes in *Babesiidae*, *Theileriidae*, *Cryptosporidiidae*, and *Sarcocystidae* (Figure 1). In the absence of conserved MSC assemblies, we searched for translational constraints that make *Plasmodium* such an unconventional parasite.

## 2. Material and Methods

### 2.1. Sequence Retrieval and Analysis

Entire proteomes, as well as individual sequences (proteins and tRNAs), were obtained directly from VEuPathDB (Eukaryotic Pathogen, Vector & Host informatics resources DataBase (accessed in 2023)) [20] (Appendix A). When protein genes were not annotated in the genomes, they were manually identified by BLAST [21]. Alignments of protein sequences were performed with M-Coffee [22] to identify and delineate additional modules (GST-like, EMAPII-like, and extra helices). Since GST-like domains have only been found in aaRS, AIMPs, and elongation factor 1 subunits to date, only these proteins have been investigated. As for tRNA sequences, when not annotated, they were identified using the tRNAScan-SE gene detection program [23] on *B. bigemina*, *T. annulata*, and *C. parvum* genomes. Eventually, tRNA sets were completed by BLAST. Box plots and line plots were generated with Excel.

### 2.2. Complex Modeling with AlphaFold2

Protein–protein complex predictions were generated in ColabFold v.1.5.3 [24] using https://colab.research.google.com/github/sokrypton/ColabFold/blob/main/AlphaFold2.ipynb (accessed on 22 November 2023) (Appendix A). Input protein sequences are those used for protein sequence alignments (Appendix A). Figures of structures were analyzed and generated using PyMOL version 2.1 [25]. Only complexes with predicted template-modeling scores (pTM) higher than 0.5 and showing potential interfaces 1, 1′, and 2 were considered (Appendix A). AlphaFold2 confidence estimates are shown for all models presented. Individual GST-like domains, all combinations of dimers, and the oligomeric GST-like backbones of MSCs were predicted using the “no template information” option in ColabFold. For *Babesia bovis*, modeling of the GST-like backbone was performed with custom templates of *Bb*-p43, ERS, and MRS obtained from predictions of heterodimers involving interfaces 1 and 2. For *Plasmodium berghei*, the Q- and M-complexes were predicted using the crystal structures *P. vivax* tRip (5ZKE), *P. berghei* ERS (8BCQ), and Raptor X models of QRS and MRS as custom templates [13]. For *Toxoplasma gondii*, dimeric GST-like heterotrimers were predicted using Raptor X [26] models of *Tg*-p43, ERS, QRS, and MRS, all of them showing the expected GST-like fold and all of them consistent with ColabFold predictions.

## 3. Results

### 3.1. Identification of GST-like Domains

The sequences of all cytosolic aaRSs and elongation factor 1 (EF1) subunits were retrieved from the nuclear genomes of the selected apicomplexan parasites (Figure 1 and Appendix A). A set of 20 cytosolic aaRSs was identified in all parasites, 4 EF1 subunits in *Plasmodiidae* and *Sarcocystidae* (EF1α, β, γ, and δ, respectively in EupathDB, corresponding to EF1A, EF1Bα, EF1Bγ, and EF1Bβ, respectively), and only 3 (EF1α, β, and γ) in *Babesiidae*, *Theileriidae*, and *Cryptosporidiidae* (Appendix A). Proteins were aligned to identify GST-like domains potentially involved in the complex formation, and their structure was predicted by ColabFold modeling. As expected, putative GST-like domains were found at the N-terminus of all EF1β, EF1γ, and AIMPs, and some of the ERSs, MRSs, and QRSs (Figure 2).

GST-like domains consist of two subdomains: the N-terminal thioredoxin-fold (β1-α1-β2-β3-β4-α2) and the C-terminal subdomain (α3 to α7) (Figure 3A). In the C-terminal domain, α-helices (α3 to α7) organize around the central helix α5, and all are parallel to each other except for α7. Helix α5 is mostly composed of hydrophobic residues and exhibits the N-capping box and hydrophobic staple motif (φ-S/T-X-X-D-φ), which is important for the stability of the fold [27] (Appendix A). However, sequence alignments and structure analysis showed that the N-capping box is missing in all *Apicomplexa* MRSs but that of *Plasmodia* (Appendix A). Several GST-like domains contained additional structures, specifically (i) the helix α8 at the C-terminus of EF1γ, ERS, MRS, and some AIMPs, (ii) the insertion of one (EF1-γ, *Plasmodiidae* MRS, and *Toxoplasma gondii* p43) or multiple α-helices (*Babesiidae*, *Theileriidae*, and *Sarcocystidae* MRS) in the loop between strands β2 and β3, and (iii) an N-terminal extension in *T. gondii* ERS (Appendix A). The GST-like domains of EF1β are exceptions as they do not contain a thioredoxin domain and are downsized to only three helices including α5 and α7 (Appendix A).

### 3.2. Description of GST-like Interactions

The oligomerization of GST-like domains involves two canonical interfaces: interface 1 and interface 2 (Figure 3B). Interface 1 allows a classical GST dimerization, where helices α2 and α3 of one monomer interact with α3 and α2 of the second monomer and are parallel to each other. The alternative interface 1′ was observed in the crystal structure of the N-terminal domain of *P. vivax* tRip [15], with helices α2 and α3 being oriented perpendicularly to each other. As for interface 2, this dimerization involves mainly a stacking interaction between two strictly conserved arginines protruding from the α7 helices of each monomer. Based on sequence analysis, the formation of interface 2 in *Apicomplexa* GST-like domains is possible only for AIMPs (except that of *Cryptosporidium*), ERSs, and the two subunits β and γ of EF1 (Figure 2 and Appendix A).

### 3.3. Pre-Tests for MSC Modeling

We set out to combine the knowledge about dimerization of GST-like domains with deep-learning-based protein structure modeling, ColabFold, to determine the structures of *Apicomplexa* MSC assemblies. For each parasite’s MSC, we modeled (i) all pairwise interactions between the different GST-like domains (Figure 4A) and (ii) the backbone of the MSC in the presence of all GST-like partners (Figure 4B). For each prediction, five models are generated, and the relative positions of the domains were explored to identify possible interfaces 1, 1′, and 2. Only the models using interfaces 1, 1′, and 2 were considered to determine the score x/5.

As a negative control, we chose to predict homodimerization in the *S. cerevisiae* AIMP Arc1p. Indeed, the crystal structure of Arc1p shows that it is a monomer, and, as expected, ColabFold modeling confirmed that no homodimers using either interface 1, 1′, or 2 could be predicted (Figure 4A). As a positive control, we chose to test the formation of EF1β and γ heterodimers, which are conserved in all eukaryotes. As expected, in all models, EF1β and EF1γ heterodimerize via their interface 2 in the absence or in the presence of EF1α and EF1δ (Appendix A), with pTM scores between 0.41 and 0.74 (Appendix A). Finally, in a previous study, ColabFold was successfully used to predict the two *Plasmodium* Q- and M-complexes, and the resulting models were highly consistent with SAXS data and mutagenesis experiments [13,16]. Based on these initial tests, ColabFold was thus considered suitable for the de novo modeling of interactions between GST-like domains and for questioning the structures of *Apicomplexa* MSCs.

### 3.4. ColabFold Predicts the Same Interaction Network in All Apicomplexa MSCs

*Theileria* and *Babesia*: Based on our sequence analysis, only AIMPs, ERS, and MRS are fused to GST-like domains in the *Theileria* and *Babesia* genera. This is reminiscent of what is known in *S. cerevisiae* where the monomeric AIMP Arc1p binds ERS and MRS [29,30]. Modeling of the dimeric interactions suggested the formation of the heterodimers AIMP:ERS in both parasites with a score of 5/5 (Figure 4A). While the AIMP:MRS heterodimer formation was observed only once in *B. bovis* (score 1/5) and not in *T. parva*, modeling of the heterotrimeric GST-like backbone (Figure 4B) resulted in the formation of AIMP:MRS subcomplexes in both *T. parva* (3/5) and *B. bovis* (5/5) MSCs. Therefore, both *B. bovis* and *T. parva* MSCs share the same organization as yeast MSC (Appendix A): AIMP (*Bb*-p43 or *Tp*-p43) binds ERS via interface 2 and MRS via interface 1. Interestingly, this evolutionary relationship is supported by the fact that *Babesia*, *Theileria*, and yeast also possess similar EF1 complexes lacking the subunit δ (Appendix A, Appendix A).

*Toxoplasma* and *Neospora*: Van Rooyen and collaborators have shown that, in *T. gondii*, five proteins associate in a single cytosolic MSC [28]. They correspond to the AIMP *Tg*-p43, ERS, MRS, QRS, and tyrosyl-tRNA synthetase (YRS). Sequence alignments showed that the *T. gondii* YRS does not contain a GST-like domain, but instead has an N-terminal α-helix (residues 1–56), strictly conserved in *N. caninum* only (Appendix A). An analysis of the pairwise interactions clearly indicated the heterodimerization of *Tg*-p43 and ERS (score 5/5) and of *Tg*-p43 and MRS (4/5) (Figure 4A). The prediction of GST-like heterotetramers completed this interaction network with QRS binding ERS via interface 1, ERS binding Tg-p43 via interface 2, and Tg-p43 binding MRS via interface 1 (Figure 4B, Appendix A). Despite being included in the modeling, the N-terminal α-helix of YRS showed no convincing interaction with any GST-like domain.

The GST-like sequences of *T. gondii* and *N. caninum* are highly conserved, suggesting that they are involved in the formation of similar MSCs. Pairwise interactions, as well as GST-like heterotetramer predictions, revealed that *N. caninum* MSC would have the same organization as *T. gondii* MSC (Figure 4 and Appendix A).

*Plasmodia*: By performing the same tests with *Plasmodium* GST-like domains—AIMP tRip, ERS, QRS, and MRS—ColabFold predicted the same interaction network as that in *T. gondii* and *N. caninum* MSCs (Figure 4B and Appendix A). However, it has been demonstrated that the formation of a single MSC containing all four proteins is not possible in *Plasmodium*. Instead, *Plasmodium* contains two ternary mutually exclusive MSCs, the Q-complex (tRip, ERS and QRS) and the M-complex (tRip, ERS and MRS) (Figure 4B, Appendix A) [13,16]. Indeed, the homodimerization of tRip in the Q-complex prevents MRS from binding to tRip (via interface 1), and the homodimerization of ERS in the M-complex prevents QRS from binding to ERS (via interface 1).

### 3.5. The Possibility of Homodimerization of Tg-p43 and Modeling of Two T. gondii MSCs

To test whether arrangements similar to those identified in the two *Plasmodium* MSCs exist in *T. gondii*, we used two copies of Tg-p43, ERS, and QRS or MRS to predict two independent dimeric heterotrimeric MSCs. None of the models proposed Tg-p43 or ERS homodimers with canonical GST-type interfaces 1 or 1′ (Figure 4B), which is consistent with the lack of prediction of canonical homodimers for *Tg*-p43 (Figure 4A). However, whether alone or in the MSC models predicted above, *Tg*-p43 homodimerized, yet always using the same non-canonical interface. This interface involves an insertion between strands β2 and β3 (Appendix A) and leads to two models consistent with the interaction network proposed in Appendix A. If biologically relevant, such a model of *Tg*-p43 homodimerization would explain not only the dimer observed in solution but would also allow the formation of a unique MSC containing all four GST-like partners as determined in [28] (Figure 4B, Appendix A).

### 3.6. Potential Consequences of tRNA Binding by MSCs

The membrane localization of both *Plasmodium* MSCs results in the presence of the tRip tRNA-binding domain (EMAPII-like) being outside the parasite and, therefore, unable to participate in the aminoacylation reaction(s) catalyzed by aaRSs within MSCs. This is different from what has been shown in the yeast complex [31,32]. However, we have proposed that the tRNA-binding domains fused to MRS (EMAPII-like domain) and QRS (positively charged helix) replace the tRip EMAPII-like domain to increase the affinity of aaRSs for their cognate tRNAs (Figure 5A). Instead, in *Toxoplasma*, immunolocalization experiments demonstrated that the MSC components are cytosolic, and the homology of *Theileria* and *Babesia* MSCs with the yeast complex (as well as a very low probability of the presence of a transmembrane helix in the corresponding AIMPs) strongly suggests that these MSCs are also cytosolic. We, therefore, propose possible roles for each of the tRNA-binding domains associated with the MSCs. *T. parva* and *B. bovis* MSCs contain only one tRNA-binding domain in the AIMP, a situation that mimics the yeast configuration. The EMAPII-like domain of the AIMP would increase the affinity of ERS and MRS for their respective tRNAs, thus increasing their aminoacylation efficiency (Figure 5B). As for *T. gondii* and *N. caninum* MSCs, they contain many additional binding domains, on AIMP, MRS, and QRS. It can be assumed that MRS and QRS tRNA-binding domains are involved in their respective aminoacylation but also in glutamylation (Figure 5C), and the AIMP EMAPII-like domain may contribute to tyrosylation.

### 3.7. What Makes Plasmodium Translation So Different That It Requires tRNA Import via Membrane-Associated MSCs?

MSC predictions indicate that, unlike *Plasmodium*, other *Apicomplexa* parasites have only one cytosolic MSC. These observations suggest that *Plasmodium* is, thus, the only *Apicomplexa* parasite with membrane-associated MSCs and which requires a tRip-directed tRNA import for optimal protein synthesis. We, therefore, inventoried the nuclear tRNA genes for each of the selected species of the *Apicomplexa* phylum (Figure 6). The nuclear genomes of *Apicomplexa* contain a relatively low number of tRNA genes compared to other protozoa (http://gtrnadb.ucsc.edu/ (accessed on 22 November 2023)). When the *P. falciparum* genome was analyzed, the authors had already noticed that this organism has few tRNA genes, with a single tRNA gene copy for a given anticodon (isoacceptor) [33,34]. We note here that these features are well-conserved not only in other *Plasmodia* but also in other *Apicomplexa* such as *Babesia*, *Cryptosporidium*, and *Theileria*, which all have between 45 and 51 tRNA genes. In these parasites, the distribution of isoacceptors is homogeneous with a few exceptions, namely, *B. bigemina*, which encodes more tRNA isodecoders than the other two *Babesia* species. Furthermore, the tRNA^Sec^ is only present in *Plasmodia*, *Neospora*, and *Toxoplasma* and is absent from the genomes of *Babesia*, *Cryptosoporidium*, and *Theileria*. This was confirmed by the joint absence of genes encoding SelB (selenocysteine-tRNA-specific elongation factor) or SelD (Selenophosphate Synthase), two proteins that belong to the specific tRNA^Sec^ machinery and allow selenocysteine insertion into proteins [35].

*Plasmodia*, *Cryptosporidium*, *Theileria*, and *Babesia* genomes contain roughly one gene copy for each tRNA isoacceptor. This is theoretically enough to support the decoding of their complete proteome. In *Toxoplasma* and *Neospora*, the number of tRNA genes is much higher, with several copies of each tRNA isoacceptor. Despite the difference in tRNA content, there is an excellent correlation (R = 0.97) between the number of nuclear tRNA genes and the size of the corresponding proteomes (total number of encoded amino acids), except for *Plasmodia* (Figure 7A). This suggests that the number of tRNA genes is not adapted to the size of the genome to be translated in *Plasmodia*.

### 3.8. Correlation between Amino Acid Usage and tRNA Availability in Apicomplexa Parasites

The sizes of the six *Plasmodia* proteomes are homogenous with an average of 3.8 +/− 0.3 M amino acids (aa). The other *Apicomplexa* genomes can be classified into two categories: (1) small proteomes corresponding to *Cryptosporidium*, *Babesia*, and *Theileria* species with sizes ranging from 1.5 to 2.8 M aa (among the three *Babesia* species, *B. bigemina* has a proteome with twice the size of the two others), and (2) large proteomes corresponding to *T. gondii* (6.7 M aa) and *N. caninum* (6 M aa) (Figure 6).

To compare the tRNAs most frequently used to synthesize proteins in *Apicomplexa* parasites, the ratio of the usage of each aa and the number of tRNA genes allowing their decoding was calculated. In Figure 7B, the calculated values represent the frequency of tRNA usage, with the highest values corresponding to the tRNAs most often recruited in gene translation for a given aa. The frequency of tRNA usage is dramatically different in *Plasmodia* and other *Apicomplexa*. The most significant differences occur in the usage of tRNA^Asp^, ^Lys^, ^Asn^, and ^Tyr^. Especially, there is an extreme usage of tRNA^Asn^ in *Plasmodia*, consistent with the presence of long asparagine homorepeats (as well as lysine homorepeats) specific to *Plasmodia* proteins (Figure 7C). However, while asparagine usage is rather homogenous in *Plasmodia* strains, this is not the case for asparagine homorepeats, that are scarcer in *P. knowlesi* and *P. vivax*.

## 4. Discussion

In eukaryotes, aaRSs and AIMPs are moonlighting proteins. They participate in tRNA charging when located in the MSC [36,37,38] and in diverse non-translational, yet crucial, functions activated by MSC release ([39,40] and reviewed in [41,42]). Among the *Apicomplexa* parasites, only the composition of *T. gondii* and *P. berghei* MSCs have been experimentally established [16,28]. While *T. gondii* has a single cytosolic MSC made up of one AIMP (*Tg*-p43) and four aaRSs (ERS, QRS, MRS, and YRSs) [28], *Plasmodium* harbors two distinct MSCs whose structures have been determined in solution [13,16]. Both *Plasmodium* complexes are constituted of a dimer of heterodimer tRip:ERS that associates with either QRS or MRS (Figure 5A). Putative MSCs were identified in all *Apicomplexa* parasites considered in this study, except for *Cryptosporidium*. They all contain one AIMP, ERS, and MRS, with or without an additional QRS. Amongst these MSC models, even those without correct interfaces, all are built on a core consisting of an AIMP:ERS heterodimer that forms via interface 2 on which MRS and/or QRS are attached (Figure 4B and Figure 5).

Several observations support the fact that the *Plasmodium* AIMP tRip, combined to one or both MSCs, is involved in the import of exogenous tRNAs inside the parasite, and that their import could compensate for the deficiency of some *Plasmodium* tRNAs. The deletion of the gene-encoding tRip not only leads to a reduction in overall protein synthesis, but also to a selective drop in asparagine-rich proteins in the parasite [14,17]. Moreover, the (human) tRNA^Asn^ is one of the favorite binders of (*P. falciparum*) tRip, which increases its chances of import [43]. However, based on the present study, this tRNA import seems to be restricted to *Plasmodia* only. This is further supported by the AIMP distance tree (Appendix A) that indicates that plasmodial tRips occupy their own orthologous group distant from the others. Moreover, the lack of resemblance between the modeled MSCs and *Plasmodium* MSCs suggests that they are neither membrane-bound nor involved in tRNA import.

*Cryptosporidium*, *Babesia*, and *Theileria* have single copies of tRNA genes which correspond to the small size of their proteomes to be produced, whereas *Toxoplasma* and *Neospora* have multiple copies of specific tRNA isoacceptors correlating with the synthesis of larger proteomes. However, this interplay does not exist in *Plasmodia*, whose tRNA genes are single copies but have larger proteomes to synthesize (Figure 6 and Figure 7A). Coding for a single copy for each tRNA isoacceptor differs from what is observed in most eukaryotes, where tRNA genes are present in multiple copies. On the one hand, the number of tRNA genes is related to the frequency of corresponding codons in the genome [44], and, on the other hand, the relative abundance of each amino acid in the proteome is not random. These rather intricate balances reflect a subtle interplay between the availability of aminoacylated tRNAs and the regulation of the expression of well-folded proteins. The correlation between amino acid usage and tRNA gene content is of particular interest in the context of parasitic organisms. *Plasmodia* amino acid usage is especially remarkable compared to other *Apicomplexa* parasites. High levels of asparagine and repeats are present in all families of *Plasmodium* proteins involved in all metabolic pathways. This extreme usage of asparagine may be explained partially by the high AT content in some *Plasmodia* genomes [45], but not exclusively. Several hypotheses have been proposed: for example, asparagine repeats act as tRNA sponges [46] or play a role in immune evasion and antigenic variation [47,48] or in protein–protein interaction [49]. Figure 7B shows that the correlation between amino acid usage and the number of tRNA genes encoded by *Plasmodia* genomes is less homogeneous than what is observed for other *Apicomplexa* and other organisms in general [44]. Indeed, the *Plasmodia* strains included in this study are characterized by more amino acids to be polymerized than the corresponding tRNAs could decode (Figure 7A). It appears that specific tRNAs, especially tRNA^Asn^, could be very limiting for the decoding of *Plasmodia* genomes (Figure 7B), independently of the presence of long asparagine homorepeats in proteins (Figure 7C) and the AT content of the genome.

## 5. Conclusions

It is accepted that changes in tRNA gene content can lead to changes in amino acid utilization, which, in turn, can lead to changes in protein structure and function [50]. In this respect, the increased utilization of a particular amino acid should lead to an increased copy number of the corresponding tRNA gene. Clearly, this is not the case in *Plasmodium* where the asparagine usage is disconnected from the number of genes encoding tRNA^Asn^. This imbalance could lead to selective pressure on proteins that use a lot of asparagine in their sequences and be beneficial in certain environments or under certain conditions. By elucidating the mechanisms underlying this relationship, we begin to better understand the evolution of *Plasmodium* genomes and pathogenicity/development. Indeed, what better way to control the production of specialized proteins than through the control of host tRNA import? The import of specific tRNAs could trigger mechanisms adapted to each stage of the parasite development, depending on the tRNA content of the different host cells [51]. We propose that the pool of tRNAs available in the host cells and eventually imported into the parasite is a kind of GPS. The import of host tRNAs would modify the concentrations of the different tRNAs present at a given time inside the parasite. Therefore, depending on the nature of the host tRNAs available and preferentially imported, the translation of one or more key regulators could vary and control the development of the parasite by indicating where it is in its developmental cycle and when to move on to the next stage.

## Figures and Tables

**Figure 1 biomolecules-14-00046-f001:**
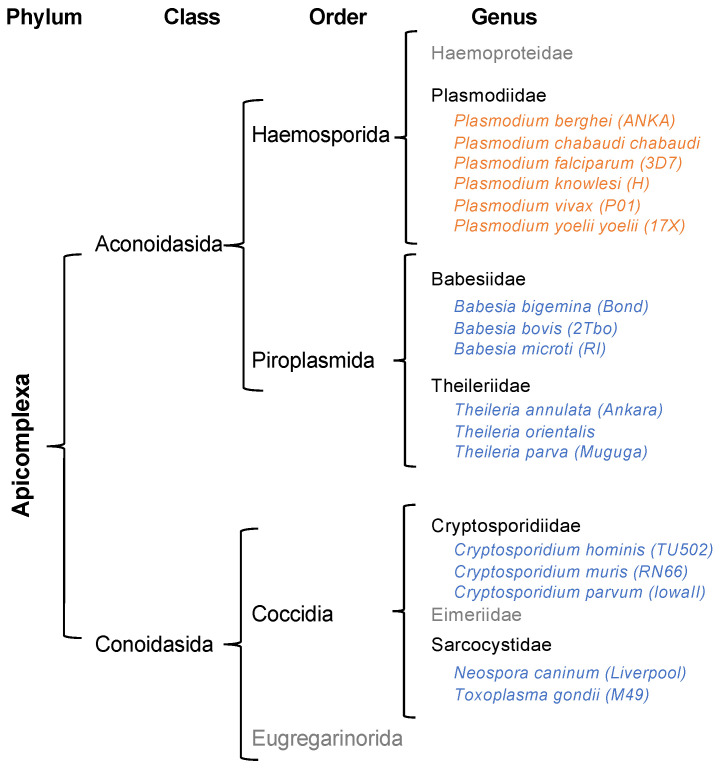
Phylogeny of *Apicomplexa* parasites. Most members of the *Apicomplexa* phylum are obligate parasites, some of which cause diseases in vertebrates. Their life cycle consists of three stages: sporozoite (infective stage), merozoite (asexual reproduction), and gametocyte (sexual reproduction). *Apicomplexa* are characterized by the presence of an apical complex responsible for the invasion of the parasite into the host cells and most of them possess an apicoplast, a plastid essential for their survival (reviewed in [18]). The phylum is divided into two classes: *Aconoidasida* and *Conoidasida*. On the one hand, the *Conoisada* include all species of *Cryptosporidium*: *C. hominis*, *C. parvum*, and *C. muris*, among others, as well as *Neospora* and *Toxoplasma*. On the other hand, the *Aconoidasida* can be classified into *Haemosporida*, consisting of *Plasmodium* species and *Piroplasmida* that include *Theileria* and *Babesia* species. This classification is the one used by the VEupathDB database (https://veupathdb.org (accessed on 22 November 2023)) [19]. *Apicopmplexa* species considered in this study are indicated in orange (*Plasmodia*) and blue (others).

**Figure 2 biomolecules-14-00046-f002:**
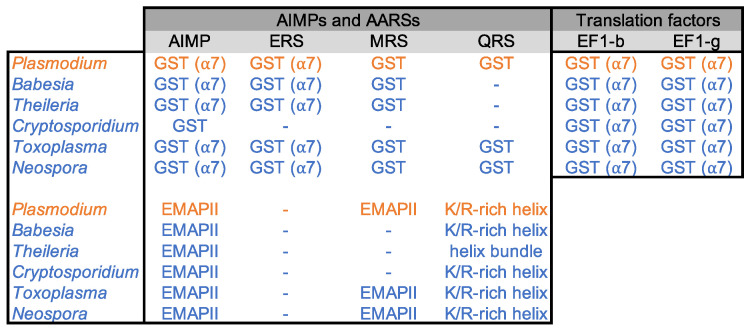
Proteins potentially involved in GST-like driven complexes in *Apicomplexa* parasites. For each *Apicomplexa* genera, homologous proteins (blue) to *Plasmodium* GST-like-containing proteins (orange) were identified by BLAST. Their additional domains located at their C-terminus are shown, as well as the presence of a putative interface 2 in the GST-like domains (indicated by α7). GST-like domains are found exclusively in eukaryotes. They are most abundant in mammals where they are found fused to EPRS, MRS, AIMP2, and AIMP3, which are part of the MSC. The GST-like domains of VRS and Ef1-γ interact, and a GST-containing CRS is produced by alternative splicing in humans and also interacts with eEF1-γ (reviewed in [12]). Other domains appended to *Apicomplexa* aaRSs have been identified and are shown in (Appendix A).

**Figure 3 biomolecules-14-00046-f003:**
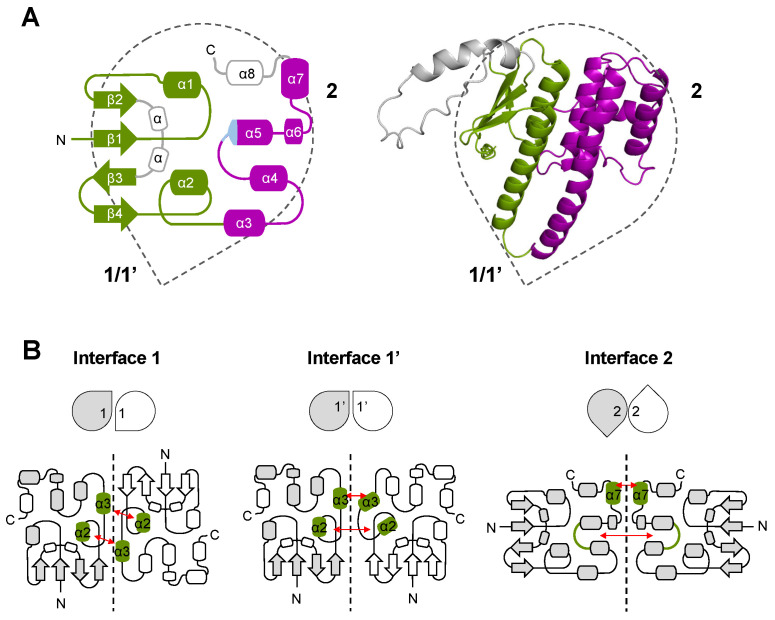
Structure and oligomerization of GST-like domains in eukaryotic MSCs and EF1. (**A**) Topological diagram and cartoon representation of a GST-like domain. The drop shape represents the orientation of the GST-like domain with oligomerization interfaces 1, 1′, and 2 highlighted. All secondary structures (α-helices and β-strands), and the N- and C-terminal ends are indicated. The thioredoxin-like subdomain (β1-α1-β2-β3-β4-α2) is colored in green, and the C-terminal helical subdomain is shown in purple (helices α3 to α7). Additional α-helices observed in some *Apicomplexa* GST-like domains are in grey. The position of the N-capping box and hydrophobic staple motif in helix α5 is colored in light blue. The model depicted in cartoon corresponds to the GST-like domain of *T. gondii* MRS predicted with ColabFold v.1.5.3 in complex with *Tg*-p43 and ERS. (**B**) Interaction interfaces involved in homo- and hetero-dimerization of GST-like domains. In each case, the drop shape representation, and the topological diagram of the GST-like dimer are shown. Interacting helices are colored in green and their contact patterns are indicated with red arrows. Interface 1 corresponds to a classical GST dimer, the two monomers being related by a 2-fold axis and interacting mainly through helices α2 and α3 in a parallel orientation. Interface 1′ is only observed in the crystal structure of *P. vivax* tRip, in which the N-termini of the two monomers are located on the same side of the homodimer and the interacting helices α2 and α3 are oriented perpendicularly. Interface 2 involves helix α7 and the loop connecting helices α4 and α5. A stacking interaction between two arginines from helices α7 is essential for dimerization and these residues are conserved only in GST-like domains from EF1-β, EF1-γ, ERS, and AIMPs.

**Figure 4 biomolecules-14-00046-f004:**
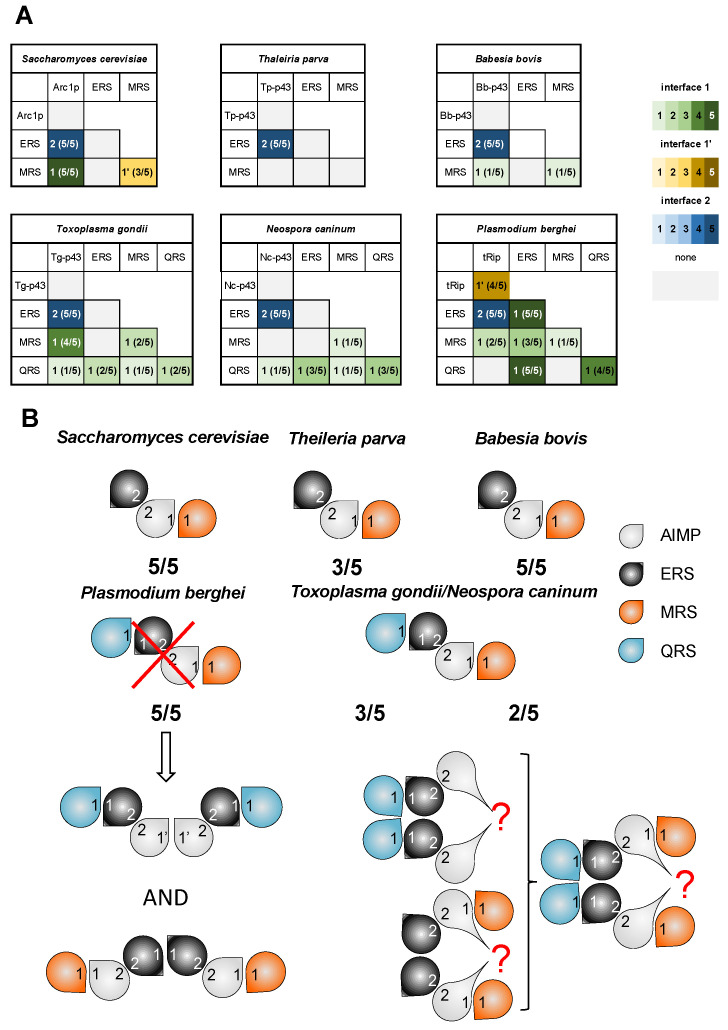
Predicted interactions between MSC GST-like domains in yeast and *Apicomplexa* parasites: (**A**) Pairwise interactions. Homo- and hetero-dimers were predicted with ColabFold using the sequences of the GST-like domains involved in yeast and *Apicomplexa* MSCs. For each combination of dimers, the occurrence of canonical GST-like interfaces 1, 1′, or 2 is indicated by a gradient of green, yellow, and blue, respectively. The score corresponds to the number of models displaying these interfaces (n = 5). (**B**) ColabFold predictions of heteromeric GST-like complexes are schematized with drop shape, with AIMP colored in grey, ERS in black, QRS in blue, and MRS in orange, and displaying interfaces 1, 1′, and 2. The *S. cerevisiae*, *T. parva*, and *B. bovis* MSCs contain 3 GST-like domains; AIMP binds MRS via interface 1 and ERS via interface 2. The 4 GST-like domains of *P. berghei, T. gondii*, and *N. caninum* share the same interaction network: AIMP and ERS heterodimerize through interface 2, QRS binds interface 1 of ERS, and MRS interface 1 of AIMP. However, it has been demonstrated that these domains form 2 independent complexes in *Plasmodium* [13,16] and that *Tg*-p43 is a dimer that belongs to a single MSC in solution [28].

**Figure 5 biomolecules-14-00046-f005:**
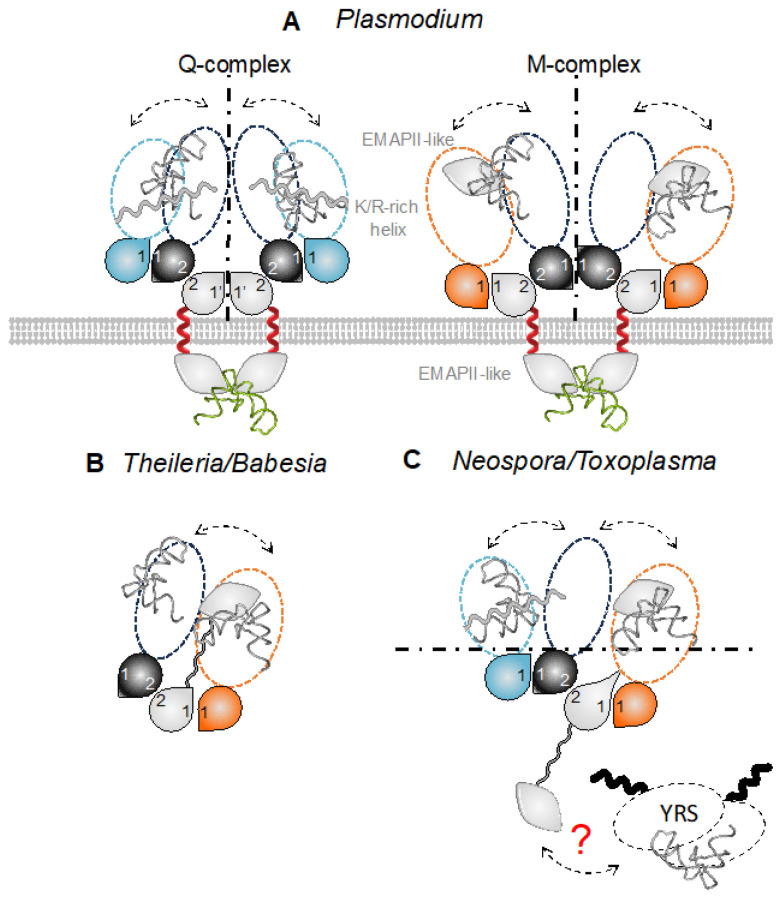
Composition and architecture of *Apicomplexa* MSC complexes. MSC models of (**A**) *Plasmodia* [13], (**B**) *Theileria* and *Babesia,* and (**C**) *Toxoplasma* and *Neospora* are shown. Color code is the same as in the legend of Figure 4. Schematic views of the complexes are built from GST-like domains (drops), minimal core enzymes (catalytic domain and the anticodon-binding domain), and additional RNA-binding domains. An EMAPII-like domain is appended to the C-terminus of MRSs (except in *Theileria* and *Babesia* complexes) and a positively charged helix is attached at the C-terminus of QRSs. These domains provide additional non-specific tRNA-binding properties to the aaRSs present in MSCs [13]. The EMAP-II like domain of AIMPs could be involved in the binding of different tRNAs, either host tRNAs in *Plasmodium* (**A**) or endogenous tRNA^Glu^ and tRNA^Met^ in *Theileria* and *Babesia* (**B**). In *Toxoplasma and Neospora*, for clarity, only one half of the MSC is shown. The YRS is a dimer with 2 N-terminal helical domains (black), but no interaction interface can be proposed. All endogenous tRNAs are now shown in grey and host tRNAs are in green.

**Figure 6 biomolecules-14-00046-f006:**
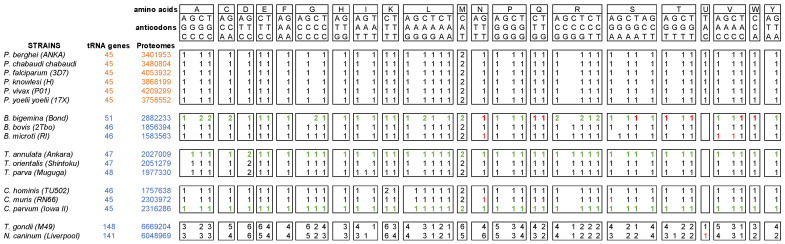
tRNA gene content in the selected *Apicomplexa* genomes. The tRNA genes were retrieved directly from annotated genomes of *Plasmodia*, *B. bovis*, *B. microti*, *T. orientalis*, *T. parva*, *C. hominis*, *C. muris*, *T. gondii*, and *N. caninum* (EupathDB). There are indicated in black. The tRNA genes identified using the tRNAscan-SE in *B. bigemina*, *T. annulata*, and *C. parvum* are shown in green. Some “missing” tRNAs in the genomes of *B. bigemina*, *B. microti*, *C. muris*, and *N. caninum* were found manually by BLAST with a sequence from an organism of the same genus, and are indicated in red. The number of tRNA genes and genome sizes are shown in orange for *Plasmodia* and in blue for other *Apicomplexa*.

**Figure 7 biomolecules-14-00046-f007:**
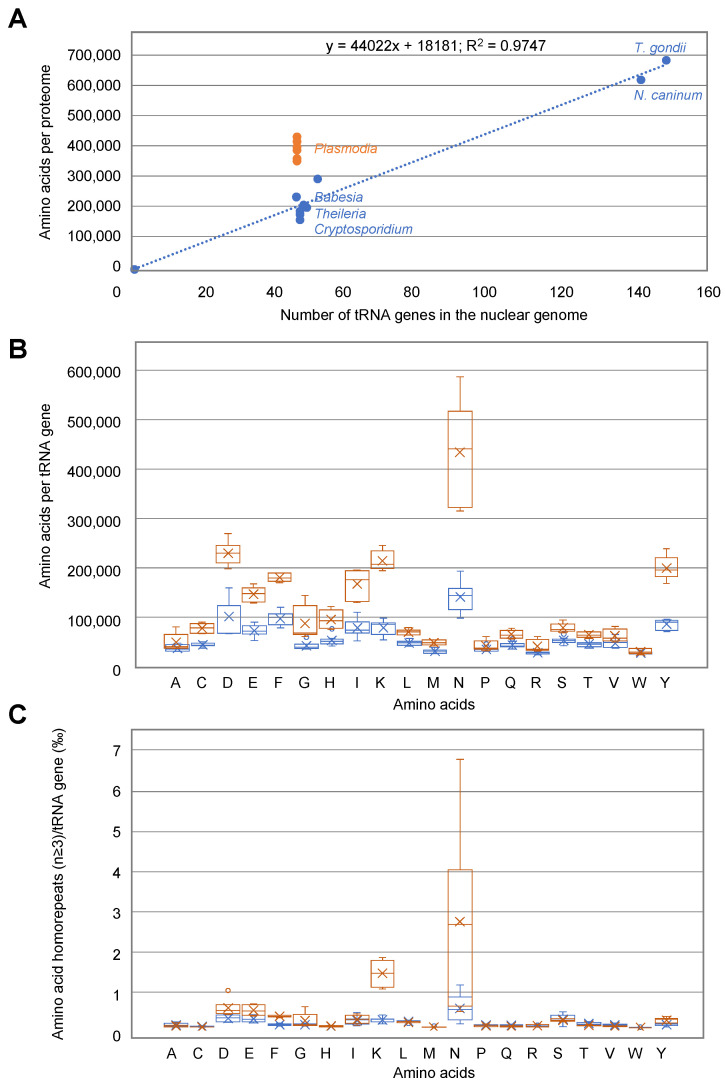
Comparison of tRNA usage in *Apicomplexa* parasites: (**A**) Correlation between proteome size and the number of tRNA genes in *Plasmodia* (orange) and other *Apicoplexa* (Blue). Data are from Figure 6. (**B**) tRNA usage displayed as amino acids decoded per tRNA gene (all isoacceptors together) in *Plasmodia* (orange) and other *Apicomplexa* parasites (blue). Some *Plasmodia* tRNAs (Asp, Ile, Lys, Asn, and Tyr) are potentially highly utilized during translation, compared to their homologues in other *Apicomplexa* parasites. (**C**) tRNA usage displayed as amino acid homorepeats decoded per tRNA gene (all isoacceptors together).

## Data Availability

Data are contained within the article and Appendix A. All ColabFold predictions are available via the following link: https://seafile.unistra.fr/d/4c9ba996bf514394ae75/ (accessed on 22 November 2023).

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
