# Peer review of "Plasmodium, the Apicomplexa Outlier When It Comes to Protein Synthesis"

_biomolecules, 2023, doi:10.3390/biom14010046_

Round 1

Reviewer 1 Report

Comments and Suggestions for Authors

Review of Ponce and Frugier, 2023

This manuscript combines bioinformatic analysis and structural models of Apicomplexa multisynthetase complex (MSC) proteins to predict the organization of MSCs in this class of parasites. Prior characterization of Plasmodium strains identified a MSC protein tRip that recruits host tRNAs to the parasitic cytoplasm MSC for aminoacylation. Other Apicomplexa representatives have some common features to their MSC organization, but Plasmodium are distinct in cellular location, multimeric organization, and need for particular tRNAs to accomplish proteome synthesis.

The rationale for the work was clearly presented, and the background was appropriate. This reviewer found it particularly helpful that the Apicomplexa phylum was schematically presented to point out where Plasmodium fits into the family.

The bioinformatic and computational methods were clearly described, including assembly of the protein and tRNA sequences from EuPathDB if previously annotated or from BLAST or tRNAScan-SE for those not previously identified. Prediction of protein complexes was performed by AlphaFold2 and ColabFold.

The authors were thorough in their comparison, including 17 Apicomplexa species across the five genera, although it seems there is perhaps a sixth genus in Figure 1 that is very light grey – is that intended to be seen? If so, it’s quite difficult to read.  

The schematic presentation of the Interface 1/1’/2 organization, including the topological maps, color coding, and teardrop shape were quite effective to communicate a sophisticated protein assembly in a clear manner. Using S. cerevisiae as a positive control for the predicted pairwise interactions is quite helpful.

The authors connect their MSC predictions with the distinctive life-cycle features of Plasmodium relative to other Apicomplexa species. In particular, the membrane-bound nature of the Plasmodium MSC (versus other MSCs that are cytoplasmic) necessitates the tRip protein facilitating host tRNA transport.  What this reviewer may have missed is whether other Apicomplexa representatives use their own tRNAs (with the GST-like domains making aminoacylation more efficient) or whether host tRNAs are also used by other Apicoplasts (see note on Figure 4/line 281). The analysis of tRNA genes and proteome size suggests that parasite-encoded tRNAs are sufficient to meet protein biosynthesis demands except in the case of Plasmodium, which is why host tRNAs are needed. But is it known whether Babesia, Cryptosporidium, and Toxoplasma genera are able to use host tRNA? This is certainly outside the scope of this work, but the analysis presented in Table 1 and Figure 5 prompts the question.

Minor points:

A couple of the images are hard to read and would benefit from larger font sizes, for example Figure 3A and Table 1.

Typos/language:

Line 60: Plasmodiidae missing an s

Figure 1: There seems to be a few very pale grey lines of text in the figure itself – if these should be read they need to be darker.

Line 80, 86, and elsewhere: I think Blast should be BLAST as it’s an acronym for Basic Local Alignment Search Tool.

Line 102: and not et.

Line 114: I would prefer “structure was predicted” not verified, since it’s a model.

Line 129: The N-capping box shown in Figure S2E is shown in blue text, but it might also help to actually draw a box around that region of sequence.

Line 196: This sentence is confusing – perhaps change to “In the S. cerevisiae…”

Line 226: “The GST-like sequences of…”

Line 229: “…organization as…” (not than)

Line 281: should the tRNAGlu or tRNAMet in Theileria and Babesia refer to parasite tRNA rather than host? I find this a bit confusing.

Line 346: “participate in tRNA charging” and “in diverse” not to.

Line 381: The sentence beginning “Especially, Plasmodia…” is confusing. Is AT content high in Plasmodium? What is remarkable about amino acid usage – is it primarily the high Asn content and homorepeats?

Comments on the Quality of English Language

The writing is clear and careful. There are just a few examples of word choice that could be corrected by journal editors.

Reviewer 2 Report

Comments and Suggestions for Authors

Review of JRJ Ponce et al., Plasmodium, the Apicomplexa outlier when it comes to protein synthesis, submitted to Biomolecules.

Summary

This fascinating and well written article by JRJ Ponce et al. describes work initially inspired by the uniqueness of Plasmodium tRip. By bioinformatic analysis the authors note that the Plasmodium proteome contains an unusually high number of asparagine residues relative to the single extant copy of Asp tRNA. The authors also use structural and binding considerations to bolster their model of what role tRip might be uniquely playing in Plasmodium, in particular giving consideration to potential interactions between GST-like domains. The authors note in their discussion that multiple lines of evidence support the possibility that Plasmodium AIMP tRip is involved in import of exogenous tRNAs. Further, they postulate that there may be an interesting reason, suggesting that perhaps “the pool of tRNAs available in the host cells and eventually imported into the parasite is a kind of GPS”, by which this parasite can better sense its external environment when coordinating its developmental program.

Given the human impact of Plasmodium, these findings could have very impactful implications going forward. The most intriguing suggested hypothesis obviously points the way to possible future directions for this work.

Specific Comments

1. “Toxoplasma gondii is a source of toxoplasmosis associated to congenital neurological birth defects (for example, encephalitis and ocular disease) [4,5];” here I might suggest “associated with” rather than “associated to”.

2. In many places in the references, within article titles necessary italicization is missing (e.g. Toxoplasma gondii). This is usually an artifact of reference management software but should be corrected in the final proofs before flattening the text to pdf.

Comments on the Quality of English Language

1. “Toxoplasma gondii is a source of toxoplasmosis associated to congenital neurological birth defects (for example, encephalitis and ocular disease) [4,5];” here I might suggest “associated with” rather than “associated to”.

Reviewer 3 Report

Comments and Suggestions for Authors

The manuscript by Jaramillo Ponce and Frugier describes the use of structural modeling to predict the assembly of multi aminoacyl-tRNA synthetase complexes (MSCs) in a variety of Apicomplexa species. In contrast to the experimentally verified, membrane-bound complexes that have been identified in Plasmodium species, the models generated suggest that MSCs in all non-Plasmodium species are cytosolic and that they involve a different set of ARS proteins. The authors also analyze amino acid content of Apicomplexa proteomes and tRNA content of the corresponding genomes to determine that Plasmodium species encode more Asn rich proteins, including many Asn repeats, despite having similar numbers of Asn tRNAs as other Apicomplexa species. This imbalance is suggested to explain the unusual, tRNA-importing, behavior of the Plasmodium MSC. 

The methods that the authors employ are generally sound, and their analysis is thorough. There are a few points in the analysis that should be made more clear and a few places where the presentation could be improved. 

• The unusual alternative MSCs in Plasmodium center around the membrane-bound protein that the authors have designated as tRIP. This protein should be evolutionarily related to aminoacyl-tRNA synthetase interacting multifunctional proteins (AIMPs) from other Apicomplexa species. A phylogenetic tree and complete sequence alignments that demonstrate this relationship (or alternatively, the lack of such a relationship) would strengthen the authors' assertion that the Plasmodium MSC is "like no other". 

• The authors assume that all protein-protein interactions will occur via GST-like domains. While this assumption seems quite valid, given the observed composition of MSCs in a variety of eukaryotes, an explicit analysis that demonstrates, for example, the lack of other unusual appended or inserted domains in Apicomplexa ARS would help justify this choice. Similarly, a schematic showing which ARSs from which species contain GST-like appendages would be helpful. (Table 1 and the text in lines 122-135 provides some of this information, but a linear schematic would be more clear)

• Detailed molecular models are presented for assemblies of EF1 in Figure S3, but only schematics are presented for the MSC complexes. At least a few representative MSC structures should be included. 

• Finally, it is surprising to me that the authors make no attempt to explain the prevalence of Asn containing proteins in Plasmodium species. Is the high prevalence of Asn general in all Plasmodium proteins or are there a few Plasmodium specific proteins that explain the difference in amino acid content? Putative functions of Asn-repeat containing proteins would be of particular interest. While a complete analysis might be beyond the scope of this work, a discussion of possibile explanations is warranted. 

Comments on the Quality of English Language

At several points in the manuscript, semicolons are used in a non-standard manner, with the first word after the semicolon capitalized. 
